# On General Language Understanding

**David Schlangen**

Computational Linguistics / Department of Linguistics
University of Potsdam, Germany
`david.schlangen@uni-potsdam.de`

## Abstract

Natural Language Processing prides itself to be an empirically-minded, if not outright empiricist field, and yet lately it seems to get itself into essentialist debates on issues of meaning and measurement ("Do Large Language Models Understand Language, And If So, How Much?"). This is not by accident: Here, as everywhere, the evidence underspecifies the understanding. As a remedy, this paper sketches the outlines of a model of understanding, which can ground questions of the adequacy of current methods of measurement of model quality. The paper makes three claims: A) That different language use situation types have different characteristics, B) That language understanding is a multifaceted phenomenon, bringing together individualistic and social processes, and C) That the choice of Understanding Indicator marks the limits of benchmarking, and the beginnings of considerations of the ethics of NLP use.

## 1   Introduction

In early 2019, Wang et al. (2019b) released the "General Language Understanding Evaluation" (GLUE) benchmark, with an updated version (Wang et al., 2019a) following only a little later. Currently, the best performing model on the updated version sits comfortably above what the authors calculated as "human performance" on the included tasks.[1] This can mean one of two things: Either General Language Understanding in machines has been realised, or these benchmarks failed to capture the breadth of what it constitutes. The consensus now seems to be that it is the latter (Srivastava et al., 2022; Liang et al., 2022).

In this paper, I try to take a step back and ask what "General Language Understanding" (GLU) implemented in machines could mean. The next

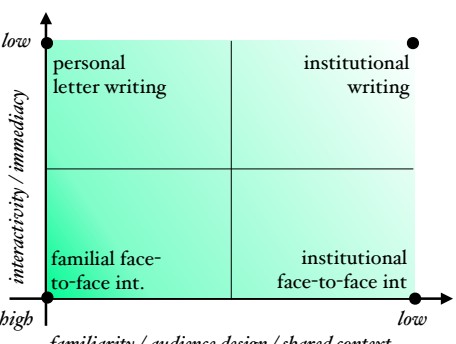

Figure 1: A Space of Language Use Situation Types

section dives into the *general* part of GLU, Section 3 into the *understanding*, as a cognitive process. Section 4 zooms out, and looks at conditions under which a *model* of GLU ceases to be *just* a model. In the course of the discussion, I will derive three desiderata for models of GLU and their evaluation.

## 2   Types of Language Use

Language can be used for many purposes (e.g., ordering dinner, teaching, making small talk with friends) and via various types of media (e.g., letters, computerised text messages, face-to-face).[2] As Fillmore (1981, p. 152) observed, one setting appears to be primary, however: "The language of face-to-face conversation is the basic and primary use of language, all others being best described in terms of their manner of deviation from that base." A detailed, multi-dimensional categorisation of these deviations can be found in (Clark and Brennan, 1991; Clark, 1996); for our purposes here, this can be collapsed into two dimensions, as in Figure 1. Along the vertical axis, we move from high interactivity as it can be found in live interaction, to low- or non-interactive language use, as it is made possible by technical mediation (via

---

[1]At 91.3, compared to the 89.8 in the paper; `https://super.gluebenchmark.com/leaderboard`, last accessed 2023-06-09.

[2]Where these purposes all come with their specific constitutive constraints on the language use, see e.g. (Bakhtin, 1986), (Wittgenstein, 1953/84, §23)).

writing or recorded messages). What are the consequences of the changes? The increase in mediation comes with a loss of immediacy, which reduces opportunities of the addressee to influence the formulation of the message, or in general to contibute. Consequently, low-immediacy use situations are appropriate more when it is one language producer who wants to convey a larger contiguous message, and not when language is used to guide collaborative action. It is reasonable to expect this difference to have an effect on the *form* of the language that is produced, and indeed this is what is typically found (Miller and Weinert, 1998; Halliday, 1989). Differences can also be found in the range of its *functions*: the range of speech acts that can be found in high-interactivity settings is larger, and includes all kinds of *interaction management acts* (Asher and Lascarides, 2003; Ginzburg, 2012; Bunt, 1996), the understanding of which requires reference to the state of the interaction. On the horizontal axis, we move from language use between speakers who share an extensive history of previous interactions and/or a rich shared situational context, to use between speakers who do not. Consequently, the kind of background information that the speakers can presuppose changes, leading to a need to make much more of the presuppositions explicit in the "low shared context" setting. This leaves us with a quadrant (top-right) where a lot of the "understanding work", at least if the language production is good, has to be "front-loaded" by the language producer, who cannot rely on the addresse intervening (bottom row) or the availability of much shared context (left column).

We can use this diagram to make several observations. First, while the ontogenetic trajectory takes the human language learner from the strongest kind of the "basic and primary" form of language—namely child/caretaker face-to-face interaction (Clark, 2003)—outwards into regions of which some are only accessible via formal education (writing in general, then technical/scientific writing), the trajectory for Natural Language Understanding in NLP takes the exact opposite direction, only now moving from the top-left corner of processing formal writing further towards the origin (Bisk et al., 2020).[3] This does not have to mean

anything, but it is worth noting—however humans do it, the fact that the more abstract types of language production found in the top-right quadrant come less easy to them may indicate that the methods that humans use to process language are taxed harder by them. (We might term the question of whether this is an essential or incidental feature the *acquisition puzzle*.)

Second, we can note that, as a consequence of this development trajectory, all of the extant large scale, "general" evaluation efforts (Srivastava et al., 2022; Liang et al., 2022) target this top-right quadrant. No standard methods have yet been proposed for evaluating models that increase interactivity and/or context dependency.[4] This might be due to the factor that an increase in context-dependence requires concrete, and hence, less general setups; but given the agenda-forming function of benchmarks, this is concerning for the emergence of a field of true GLU. (We might term this the *coverage problem*.) We derive from this discussion the first desideratum.

***Desideratum 1: Models of "General Language Understanding" must be general also with respect to language use situation types, and must cover situated as well as abstracted language use.***

## 3 Understanding as a Cognitive Process: Inside the Understander

The previous section looked at generality in terms of coverage of language use situations. This section will look at one aspect of the *understanding* part in "General Language Understanding". What is understanding? The classic view in NLP is well represented by this quote from a seminal textbook: "*[the understanding system] must compute some representation of the information that can be used for later inference*" (Allen, 1995, p.4).

Taking up this "*actionable representation*" view, and at first focussing on "text understanding", Figure 2 (left column) shows an attempt to compile out of the vast literature on language understanding, both from NLP, but also from linguistics and psycholinguistics, a general (if very schematic) picture—a picture that at this level of detail would not be incomprehensible to the contemporary reader of Allen (1995). The model assumes that the language understander possesses a model

---

[3]Note that while there is renewed interest in "embodied" language use in NLP (Duan et al., 2022; Gu et al., 2022), outside of the small interactions with the neighbouring field of social robotics, there is little work on actual embodiment that could lead to models of "face-to-face" interaction.

[4]For evidence that the increase in interactivity is inconsistent, see (Doğruöz and Skantze, 2021). A theoretical proposal for an evaluation method is given by (Schlangen, 2023), with a first realisation attempted by (Chalamalasetti et al., 2023).

of the language in which the material that is to be understood is formulated; here in the more narrow sense that it is a model of a mapping between form and meaning (representation), roughly of the *scope* aimed at by the formalisations such as those of Chomsky (1957) or Pollard and Sag (1994).[5] This model interfaces with world knowledge at least in the lexicon (Pustejovsky and Batiukova, 2019), via knowledge of *concepts* (Murphy, 2002; Margolis and Laurence, 2015). The *world model*, however, in this view more generally needs also to contain "common sense knowledge" about the workings of the world (e.g., as script knowledge, common sense physics, etc.; Allen and Litman (1990); Nunberg (1987)).

The central representation however in this model is the *situation model* representing the described situation, in the broadest sense (which may or may not be congruent with the reporting situation; Johnson-Laird (1983); van Dijk and Kintsch (1983)). To give an example, Winograd schema sentences (Levesque et al., 2012) such as (1) (taken from (Wang et al., 2019a)) can in this scheme be understood as inducing a situation model, for which *language* and *world model* suggest a preferred understanding (namely, that it was the table, as the patient of the breaking event, that was built out of the fragile material).

(1)    The large ball crashed right through the table because it was made of styrofoam.

To be able to separate between elements of the situation model that may have been implied and those that have been explicitly mentioned, a representation of the discourse is required (Heim, 1983; Kamp, 1981). Its structure moreover can constrain what can be inferred, as in (2-a), where 'car' is not available as antecedent for the pronoun.

(2)    a.    The nearly bankrupt company did not own a car. It was on the verge of collapse.
       b.    The nearly bankrupt company did own a car. It was on the verge of collapse.

(We will skip over the agent model for now.)

That "language understanding" is internally structured and draws on various types of knowledge is implicitly acknowledged also in modern attempts at evaluating the performance of NLU

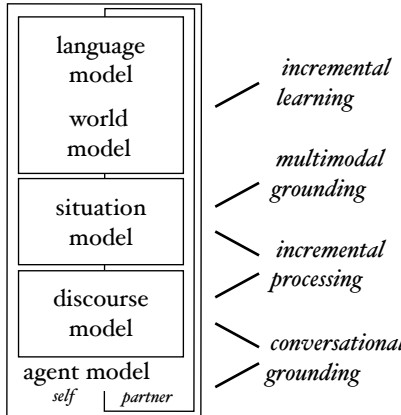

Figure 2: A Model of Understanding as a Cognitive Process

models, for example in the diagnostic dataset included in SuperGLUE (Wang et al., 2019a), or in the checklists of Ribeiro et al. (2020). This assumption also underlies the fertile field of representation probing (e.g., (Marvin and Linzen, 2018; Belinkov, 2022; Loáiciga et al., 2022; Schuster and Linzen, 2022)), which tests for mapping between such theoretically motivated assumptions and empirical findings on processing models. However, the underlying assumptions are rarely made explicit, not even to the degree that it is done here (but see (Trott et al., 2020; Dunietz et al., 2020))— which, I want to claim here, should be done, in the interest of *construct validity* of measurement (Flake and Fried, 2020).

But we are not done. What I have described so far may capture *text understanding*, but once we move outwards from the top-right quadrant of Figure 1, the collaborative nature of interaction, and with it the importance of the *agent model*; and in general the processual nature, and with it the importance of the various *anchoring processes* shown in Figure 2 on the right, come into focus. In order: Where it might be possible to understand text, particularly of the de-contextualised kind described above, without reference to its author, the further one moves towards the origin of the language use space (Figure 1), the clearer it becomes that the understander needs to represent its beliefs about the interacting agent. This is indicated by the *agent model* in Figure 2, where the segment for the partner contains information about which parts of the other models the understander deems to be *shared* (Bratman, 1987; Cohen et al., 1990). The model of Clark (1996) makes building up this *common ground* the point of understanding, and the process of managing this common ground via interaction,

---

[5]Which is not to imply that an implemented language understanding system should be *based on* such formalisations.

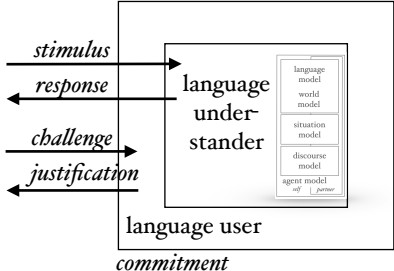

Figure 3: A Model of Understanding as a Social Process

*conversational grounding*, its central element. In conversational grounding, not only processes of *repair* (asking for clarification) are subsumed, but also "positive" indicators of understanding (such as producing a relevant continuation). This process is made possible by the fact that processing of material happens, very much unlike the current assumptions in NLP, in an incremental fashion (Levinson, 2010; Christiansen and Chater, 2016; Schlangen and Skantze, 2009), allowing for timely adaptations and interventions. Another natural phenomenon in interaction is covered by the process of *incremental learning* (Hoppit and Laland, 2013; Harris, 2015): If, in the course of an interaction, I am introduced to a fact previously unknown to me, and I accept it through conversational grounding, I am expected to be able to later draw on it. The final process is the only one that has seen some attention recently in NLP, *multimodal grounding*; which here however is meant not just to cover the word-world relation (Harnad, 1990; Chandu et al., 2021), but also the grounding of meaning-making, in face-to-face situations, in multimodal signals from the speaker (Holler and Levinson, 2019; McNeill, 1992; Kendon, 2004).

The takeaway from this shall be our second desideratum.

**Desideratum 2: Attempts at measuring performance in "General Language Understanding" must be clear about their assumptions about the underlying construct.**

(Where the model sketched in this section provides one example of how to be explicit about such assumptions.)

## 4 Understanding as a Social Process: The Understanding Indicator

The discussion from the previous section suggests a picture where a language understanding system receives a stimulus and delivers a response, which we take as an indicator of understanding. And this is indeed how typical evaluation of such a system works: The response is compared to the known, expected response, and the assumed quality of the model is a function of this comparison. This, however, is not how understanding in real use situations works: Here, we do not care about *understanding as symptom* (reflecting an inner state), but rather as *understanding as signal* (offering a social commitment). In most use situations, "computer says no" (Britain, 2004) is not good enough (or at least, should not be good enough): We want to know *why* it says this, and we want to know who takes responsibility if the reasons are found to be not good enough.[6] In other words, and as indicated in Figure 3, in this view, the understanding indicator is embedded in practices of receiving challenges and providing justifications (Toulmin, 2003 [1958]), as well as making commitments (Brandom, 1998; Lascarides and Asher, 2009); in other words, it underlies the constraints holding for the speech act of *assertion* (Goldberg, 2015; Williamson, 2000).

I can only scratch the surface of this discussion here, to make a few notes: A) The target of "normal" evaluation—improving the reliability of the understanding symptoms—certainly stands in some relation to improving the quality of the understanding signal, but it is not entirely straightforward to see what this relation is, and what its limits are. B) While the process of giving justifications when challenged may be within the range of "normal" work in NLP, and indeed is addresses by some work in "explainable AI" (Miller, 2019), whether the notion of *commitment* can ever be abstracted away from human involvement is more than questionable. C) There is a long tradition of work making similar points, coming to them from a different angle (e.g., *inter alia*, Bender et al. (2021)). I just note here that considerations originating from the philosophy of language about how meaning is underwritten by the "game of giving and asking for reasons" (Sellars, 1956) only strengthen these concerns.

**Desideratum 3: Uses of models of Language Understanding must be clear about their understanding of the Understanding Indicator, and how it is warranted.**

---

[6]A right which famously recent EU regulation (Regulation, 2016, recital 71, "right to explanation") indeed codifies, at least in principle.

# 5 Conclusions

This short paper is an invitation to debate what the meaning of "general language understanding" (in machines) could, and ought to, be. Ultimately, it may be that the answer to "can large language models *model* language understanding" is *yes*, while the answer to "can large language models understand language" has to be *no*.

## Limitations

This paper does not report on any empirical work. Is it hence out of place at a conference on "empirical methods"? I would argue that it is not, as, in the words of Dewey (1960, p.85) "all experiment involves regulated activity directed by ideas". To not be empty, empirical methods must be guided by theoretical considerations, and it is to this that this paper wants to contribute.

A limitation might be that this text was written with the thought that language understanding by machines is done for humans, and that thus the human-likeness of the understanding is crucial, because only it guarantees that generalisations go in expected directions. The thoughts developed here might not apply if the goal is to evolve machine communication that only superficially resembles natural language.

## Ethics Statement

While the paper discusses some possible limits to language understanding by machines, it does not per se question whether "general language understanding" by machines is a worthwhile, ethical goal. This should be discussed; for now, elsewhere.

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
