# OpenReview forum: "On General Language Understanding"
_EMNLP/2023/Conference — EMNLP 2023 Findings_

### Official Review · Reviewer_kY1L · 2023-07-31

**Typos Grammar Style And Presentation Improvements:** l.227
**Soundness:** 4

**Excitement:**

4: Strong: This paper deepens the understanding of some phenomenon or lowers the barriers to an existing research direction.

**Paper Topic And Main Contributions:**

This position paper tackles the question of the extent to which language models can be said to ‘understand’ language.
The author provides an analysis of general language inderstanding on two dimensions: interactivity and familiarity. They propose that, placed on this matrix, the forms of communication most natural and easy for humans are quite different from those most easily evaluated in language models.
The author proposes three desiderata for modelling understanding in language models and concludes that while such modelling is possible, it cannot be said that language models display ‘understanding’.

**Questions For The Authors:**

Question A:
Could deseridatum 2 be rewritten to be standalone rather than referring to “a model such as sketched here”? This seems a bit clunky to me.

**Reasons To Accept:**

This is an extremely interesting paper on a timely topic as there has recently been much discussion—both among researchers and the general public/media/policy makers—of the extent to which language models ‘understand’ language.

The paper is very well structured, written, and argued.

**Reasons To Reject:**

I agree with the author’s self-assessment in the ‘Limitations’ section that this work belongs at a venue like EMNLP, and see no reasons to reject the paper.

**Reproducibility:**

N/A: Doesn't apply, since the paper does not include empirical results.

**Reviewer Confidence:**

4: Quite sure. I tried to check the important points carefully. It's unlikely, though conceivable, that I missed something that should affect my ratings.

---

> ### Author Rebuttal · Authors · 2023-08-28
>
> I thank reviewer kY1L for their encouraging reaction to the paper and their willingness to see relevance for our joint scientific undertaking in the field of NLP.

---

### Official Review · Reviewer_tfyP · 2023-08-03

**Soundness:** 4

**Excitement:**

4: Strong: This paper deepens the understanding of some phenomenon or lowers the barriers to an existing research direction.

**Paper Topic And Main Contributions:**

The paper contributes theoretical thought regarding the empirical method of using machinery to understand natural language. The author's position on "General Language Understanding (GLU)", in my opinion, closely relates to Artificial General Intelligence (AGI), an area concerned with deeper understanding by implementing human-like AI. The author proposes an outline of GLU issues to be addressed and invites the community to debate those issues. The outline deals with three aspects termed "types [or purpose] of language use", "understanding as a cognitive process", and "understanding as a social process".

**Reasons To Accept:**

The claim, to present the above outline in a short paper, is justified by surveying extensive literature, including examples from the Winograd Schema collection (of difficult text for testing AI to see how human-like it can be). Language understanding quality is an apparent bottleneck of currently predominant big-data language models. Consequently, the proposed outline can, in my opinion, draw more community attention to the following: language use in conversation and language acquisition; ways the language structure can convey the role in a situation, and common sense reasoning based upon those roles; transparency of machine reasoning. In other words, the paper can draw attention to what is apparently missing from the current NLP, but could remedy its bottleneck problem.

**Reasons To Reject:**

The information in the paper often comes in quintessence chunks. I mean that the paper often summarizes, say, some extensive research in a short phrase or sentence. To make the most of it, the reader needs to be well familiar with all the works summarized, otherwise it is hard to follow. I, for one, would befit more from straightforward explanations and technical examples of how the discussed understanding is implementable in real life. Considering the paper format and claim, my personal benefit comments have more to do with "Excitement".

**Reproducibility:**

N/A: Doesn't apply, since the paper does not include empirical results.

**Reviewer Confidence:**

4: Quite sure. I tried to check the important points carefully. It's unlikely, though conceivable, that I missed something that should affect my ratings.

---

> ### Author Rebuttal · Authors · 2023-08-28
>
> I thank reviewer tfyP for their receptive and encouraging comments and for their willingness to engage with the argument given here.
>
> I agree with the comment on the stark compression of the argumentation and the references to the literature. Should the paper be accepted, I will use the additional page to decompress the text a bit and add illustrative examples.

---

### Official Review · Reviewer_uYnW · 2023-08-05

**Soundness:** 1

**Excitement:**

2: Mediocre: This paper makes marginal contributions (vs non-contemporaneous work), so I would rather not see it in the conference.

**Missing References:**

H. J. Briegel. On creative machines and the physical origins of freedom. Scientific Reports, 2:522, 2012.

J. Pearl. Causality: Models, Reasoning and Inference. Cambridge University Press, Cambridge, 2000.

J. Pearl. The seven tools of causal inference, with reflections on machine learning. Communications of the ACM, 62(3), 2019.

J. Pearl and D. Mackenzie. The Book of Why. Basic Books, New York, 2018.

K. Ried, B. Eva, T. Mueller, and H. J. Briegel. How a minimal learning agent can infer the existence of unobserved variables in
a complex environment. CoRR, abs/1910.06985196, 2019.

**Paper Topic And Main Contributions:**

This paper takes a philosophical stance and aims to develop a model of General Language Understanding by which LLMs can be evaluated.

**Questions For The Authors:**

None.

**Reasons To Accept:**

N/A

**Reasons To Reject:**

The contribution seems misplaced in a conference on Empirical Methods.  Though this problem is addressed proactively as part of the paper, making the point that empirical enterprise also needs background thinking, the argumentation in the paper is not convincing.  The paper reviews classics of symbolic AI and their claims (nothing new here) but does not address literature that is relevant particularly with respect to sub-symbolic AI.  This includes Pearl's Ladder of Causation as well as philosophical thoughts about machine agency, as pursued for example, by Thomas Mueller.

**Reproducibility:**

N/A: Doesn't apply, since the paper does not include empirical results.

**Reviewer Confidence:**

4: Quite sure. I tried to check the important points carefully. It's unlikely, though conceivable, that I missed something that should affect my ratings.

---

> ### Author Rebuttal · Authors · 2023-08-28
>
> There's probably not much I can do to change reviewer uYnW's mind. I shall nevertheless try to address some of the issues raised.
>
> > The contribution seems misplaced in a conference on Empirical Methods. Though this problem is addressed proactively as part of the paper, making the point that empirical enterprise also needs background thinking, the argumentation in the paper is not convincing.
>
> There is an argument in the paper that foundational considerations do belong in a conference on empirical methods, especially if -- and this diagnosis is implied by the paper, and seems fairly common in the field -- the enthusiastic application of these methods has lead to a crisis of understanding, as evidenced by the discussion (also carried forward at EMNLP) of whether our empirical methods are aiming at and measuring the right things. Is this what uYnW is disagreeing with?
>
> There is also an argumentation in the paper about how to remedy this. I would welcome engagement with this argumentation, to see where it is deemed unconvincing.
>
>
> > The paper reviews classics of symbolic AI and their claims (nothing new here)
>
> Can I respectfully point out that most of the cited papers are from linguistics, not "symbolic AI"? Furthermore, I would also like to stress -- and thank the reviewer for pointing out the need to clarify this yet more in the paper -- that the argument is for clarity on the underlying construct, the *what*, and is not aiming to provide strictures on the *how* of modelling the phenomenon. That is, the contrast between "symbolic" and "sub-symbolic AI" is orthogonal to the argument of the paper. The analytical observations of linguistics have not become irrelevant, just because the computational modelling methods may have changed.
>
>
> > does not address literature that is relevant particularly with respect to sub-symbolic AI. This includes Pearl's Ladder of Causation as well as philosophical thoughts about machine agency, as pursued for example, by Thomas Mueller.
>
> Thank you for these interesting pointers to additional literature, some of which I indeed had not seen before. Again, this literature seems to be more about *how* to model, and not about *what* to measure. It seems to me that reviewer uYnW thinks that they have a better proposal. This may very well be! I would like to see that proposal being given the same room to be developed and presented as I am asking to be given here.

---

### Meta-Review · Area_Chair_aov1 · 2023-09-18

**Recommendation:** 3

**Metareview:**

This paper presents a theoretical contribution on the highly relevant and timely topic of natural language understanding, in the light of recent technological developments. A few pointers to the literature have been suggested to better position the work in the context of EMNLP. Some definitions and theoretical principles may be further expanded to improve the readability and self-containment of the paper.
While different from the "usual" contribution, this short paper has the potential to spark interesting and important discussions.

---

### Decision · Program_Chairs · 2023-10-07

**Decision:**

Accept-Findings

**Comment:**

This paper presents a theoretical contribution on the highly relevant and timely topic of natural language understanding, in the light of recent technological developments. A few pointers to the literature have been suggested to better position the work in the context of EMNLP. Some definitions and theoretical principles may be further expanded to improve the readability and self-containment of the paper.
While different from the "usual" contribution, this short paper has the potential to spark interesting and important discussions.